# On the Development of Material Constitutive Model for 45CrNiMoVA Ultra-High-Strength Steel

**Xin Hu [1], Lijing Xie [1,\*], Feinong Gao [1] and Junfeng Xiang [2]**

[1] School of Mechanical Engineering, Beijing Institute of Technology, Beijing 100081, China; 13810438099@163.com (X.H.); feinong_gao@163.com (F.G.)

[2] School of Electromechanical and Automotive Engineering, Yantai University, Yantai 264005, China; xiang_junfeng@126.com

\* Correspondence: rita_xie@bit.edu.cn; Tel.: +86-10-6891-2716

**Abstract:** For the implementation of simulations for large plastic deformation processes such as cutting and impact, the development of the constitutive models for describing accurately the dynamic plasticity and damage behaviors of materials plays a crucial role in the improvement of simulation accuracy. This paper focuses on the dynamic behaviors of 45CrNiMoVA ultra-high-strength torsion bar steel. According to investigation of the Split-Hopkinson pressure bar (SHPB) and Split-Hopkinson tensile bar (SHTB) tests at different strain rate and different temperatures, 45CrNiMoVA ultra-high-strength steel is characterized by strain hardening, strain-rate hardening and thermal softening effects. Based on the analysis on the mechanism of the experimental results and the limitation of classic Johnson-Cook (J-C) constitutive model, a modified J-C model by considering the phase transition at high temperature is established. The multi-objective optimization fitting method was used for fitting model parameters. Compared with the classic J-C constitutive model, the fitting accuracy of the modified J-C model significantly improved. In addition, finite element simulations for SHPB and SHTB based on the modified J-C model are conducted. The SHPB stress-strain curves and the fracture morphology of SHTB samples from simulations are in good agreement with those from tests.

**Keywords:** dynamic behaviors; modified J-C model; J-C damage model; phase change; finite element simulation

## 1. Introduction

45CrNiMoVA alloy is a type of low-alloy ultra-high-strength steel, widely used in aircraft engine crankshafts, landing gears and other high-strength structural parts and torsion bars because of its high strength, high hardenability, high specific strength and other excellent mechanical properties; yet this type of steels are difficult to manufacture by welding, forming or machining [1]. The material constitutive models for describing plasticity and damage behaviors are key for the development of a finite element simulation of large plastic deformation processes [2]. Besides chemical composition, the mechanical properties of the material are influenced by material processing history such as heat treatment, so it is necessary to determine the material constitutive models based on the material mechanical tests.

Among the material constitutive models, Johnson-Cook (J-C) model is the most widely used empirical relationship for calculating the flow stress. It descripts the relationship of flow stress to plastic strain, strain rate and temperature, and is expressed as [3]:

$$\sigma = (A + B\varepsilon_P{}^n) \times \left(1 + C\ln\left(\frac{\dot{\varepsilon}}{\dot{\varepsilon}_0}\right)\right) \times \left(1 - (T^*)^m\right) \qquad (1)$$

$$T^* = \begin{cases} 0; & T < T_{\text{room}} \\ \frac{T - T_{\text{room}}}{T_{\text{melt}} - T_{\text{room}}}; & T_{\text{room}} < T < T_{\text{melt}} \\ 1; & T_{\text{melt}} < T \end{cases} \tag{2}$$

Therein, *A, B, C, n, m* are the material constants to be identified by fitting flow stress-strain data from material mechanical tests, such as quasi-static and Split-Hopkinson pressure bar (SHPB) compression tests at varying strain rate and temperature. *A, B, n* are strain related and C strain rate related. *m* reflects the material's thermal softening effect.

Material mechanical tests have their limitations at the range of strain, strain rate and temperature. The range of strain, strain rate and temperature covered by the material mechanical tests for fitting constitutive model determines the valid range in application. At present, most dynamic behaviors of materials are investigated based on SHPB tests, but it is difficult for SHPB tests to reach such high strain rate of $10^5$–$10^8$ s$^{-1}$ as in metal cutting. Several researchers have focused to develop material constitutive model for metal cutting. Tounsi [4] presented a method for identifying the material constants of the constitutive models based on a series of orthogonal cutting experiments and tested certain cutting process variables. Empirical equations are used to calculate the strain, strain rate and temperature for fitting the J-C model.

Dabboussi [5] studied the development and application of constitutive models. The J-C plasticity model and certain simplified failure model parameters for aluminum alloy, titanium alloy and stainless steel are determined based on quasi-static and SHPB compression tests. According to the comparison with the ABAQUS/Explicit simulation results, it is assumed that the stainless steel material with anisotropic properties is not suitable material for these two models due to their inherent limitations. So it is necessary to consider not only strain, strain rate and temperature, but also the material's special loading conditions in the development of the constitutive models. Therefore, some modified J-C constitutive models considering adiabatic temperature rise, extremely high strain rate, and evolution of dislocations have been established accordingly [6–8].

The constitutive models mentioned above are all based on J-C model, and the parameter fitting are all performed with least square method by making sure only one single variable such as strain, strain rate or temperature change in every round of parameter fitting, so the multi-variable coupling effects on the flow stress are ignored. In this case, the problems of fitting possibility and accuracy for a set of given data was discussed. Xiang [9,10] pointed out its applicability limitation and low accuracy and proposed a weighted multi-objective strategy for identifying material constitutive model parameters based on the Gaussian-distributed noise evaluation on the experimental data under different loading conditions. The fitting accuracy of the J-C model parameters is improved by this method and the 3D drilling simulations performed by using this set of J-C parameters is in good agreement with the tests.

In order to investigate the dynamic behavior of 45CrNiMoVA ultra-high-strength steel in large plastic deformation process, SHPB tests at different strain rate and temperature, and quasi-static compression tests at the strain rate of $10^{-4}$ s$^{-1}$ and $10^{-3}$ s$^{-1}$ are performed, and then the material constitutive model of plasticity are identified based on the weighted multi-objective fitting strategy.

In addition, quasi-static tensile tests at different stress triaxiality and the Split-Hopkinson Tensile Bar (SHTB) tests ranged from low strain-rate to medium strain-rate, low temperature to extremely high temperature are carried out. Then the J-C damage model is determined by data fitting with linear regression method.

## 2. Materials and Experimental Procedures

### 2.1. 45CrNiMoVA Steel

The investigated 45CrNiMoVA low carbon alloy steel has experienced 2.5-h annealing at 920 °C, 1.5-h quenching at 780 °C and 4-h low temperature tempering at 200 °C. Its chemical composition, physical and mechanical properties are shown in Tables 1 and 2. By observing the surface perpendicular to the rolling direction (RD) of a 10 × 10 × 10 sample, the electron backscattered diffraction (EBSD)

analysis in Figure 1 shows that the microstructure of this material includes 93% tempered martensite and 7% retained austenite. The crystal orientations of the material are shown in Figure 1d. In addition, between grains the high-angle grain boundaries (more than 15°) account for 67.93% and the small-angle grain boundaries (less than 15°) account for 32.07%, and the grain size distribution ranges are shown in Table 3. As such, this material is characterized by large number of large-angle grain boundaries and fine grains, which help improve strength and toughness [11,12].

**Table 1.** Chemical composition of 45CrNiMoVA (wt. %).

| C | Cr | Ni | Mo | V | Si | Mn |
|---|----|----|----|----|----|----|
| 0.42–0.49 | 0.8–1.1 | 1.3–1.8 | 0.2–0.3 | 0.10–0.20 | 0.17–0.37 | 0.5–0.8 |

**Table 2.** Physical and mechanical properties of 45CrNiMoVA used for simulation.

| Notation | Material Properties | Value |
|----------|---------------------|-------|
| $\rho$ | Density (kg/m$^3$) | 7800 [13] |
| $T_{melt}$ | Melting point (°C) | 1550 |
| $T_{room}$ | Room/reference temperature (°C) | 25 |
| $E$ | Elastic modulus (GPa) | 212 |
| $\nu$ | Poisson's ratio | 0.29 |
| $\dot{\varepsilon}_0$ | Reference strain rate | 0.001 |
| C | Specific heat (J/kg·K$^{-1}$) | 460 [13] |
| $\alpha$ | Thermal expansion coefficient (10$^{-6}$ K$^{-1}$) | 11.7 [13] |
| $\kappa$ | Thermal conductivity (W/m·K$^{-1}$) | 19 |

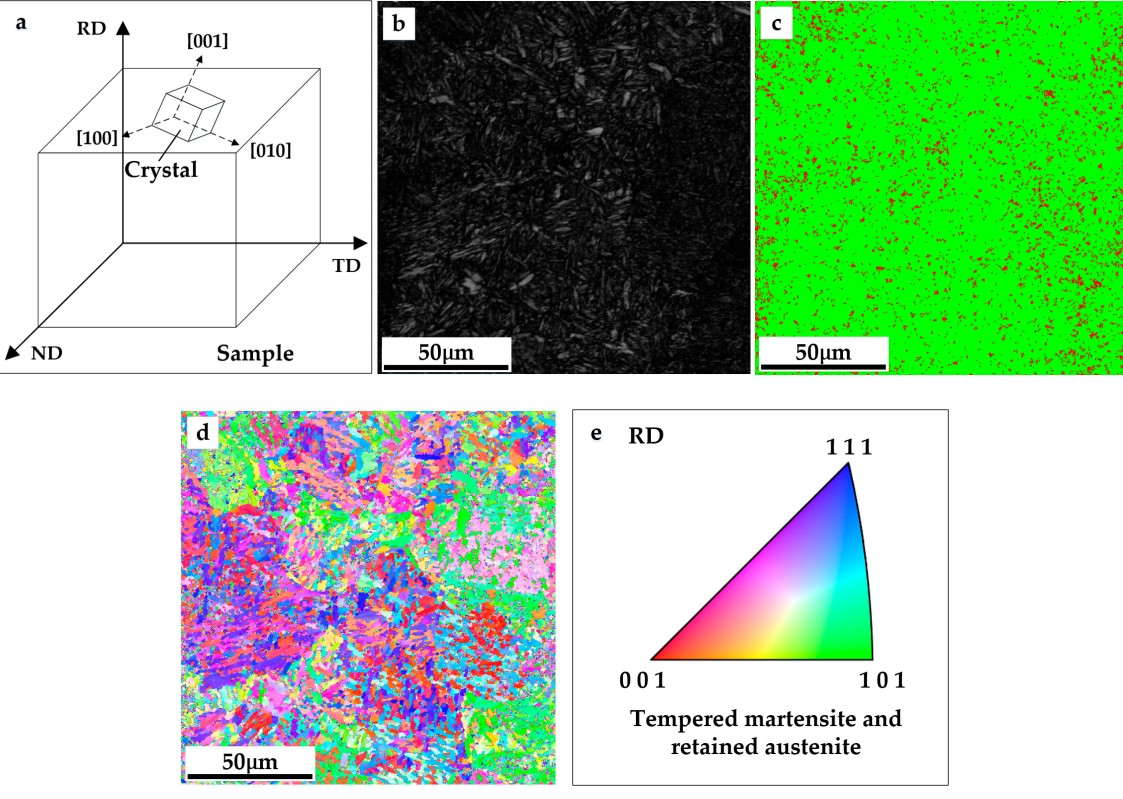

**Figure 1.** (**a**) Sample directions: RD represents rolling-direction, TD represents transverse-direction and ND represents normal-direction; (**b**) Scanning electron microscope (SEM) metallographic graph; (**c**) distribution of martensite and austenite grains, green area represents tempered martensite and red area represents retained austenite; (**d**) RD inverse pole figure; (**e**) color code map of the RD inverse pole figure.

**Table 3.** Grain size of 45CrNiMoVA.

| Grain Diameter (μm) | Fraction (%) |
|:---:|:---:|
| 0–4 | 67.52 |
| 4–8 | 27.78 |
| 8–12 | 3.31 |
| >12 | 1.39 |

*2.2. An Improved SHPB and SHTB Tests at High Temperature*

Both the SHPB tests and the SHTB tests are carried out. In the traditional high-temperature dynamic tensile and compression tests, the samples are tightly connected to the incident bar and the transmission bar by extrusion, bonding or threaded connection. Thus, connection methods will cause the end of two loading bars in contact with the sample heat up as the sample being heated, which results in the existence of temperature gradient in the loading bars. The temperature gradient leads to the decrease of Young's modulus and the change of wave velocity in the bars, these are all reasons for measurement errors in the experimental results [14–16].

Different from the traditional dynamic tensile and compression test devices, this set of test devices is equipped with a set of synchronization devices to reduce the error caused by the temperature gradient [15]. As shown in Figures 2 and 3, the sample is placed in the clamping device or on the guide rail so that it has no contact with two loading bars. After the sample is heated in the furnace to the required temperature and kept for five minutes, the synchronous device works. In the SHPB test of Figure 2, the synchronous device pushes the transmission bar to make the sample completely in contact with the two loading bars. In the SHTB test of Figure 3, the pneumatic device I pushes the sample into the clamping position between the incident bar and transmission bar, then the pneumatic device II pulls the transmission bar to eliminate the gaps between the sample and the loading bars. Finally, the strike bar or the tube is driven by the air chamber and hits the incident bar. In this way, an incident wave is generated and transmits through the sample and the bars. The time interval from the hot sample coming in contact with the cold-loading bars to hitting is controlled within 50 ms to reduce the temperature gradient effect to a minimum [17].

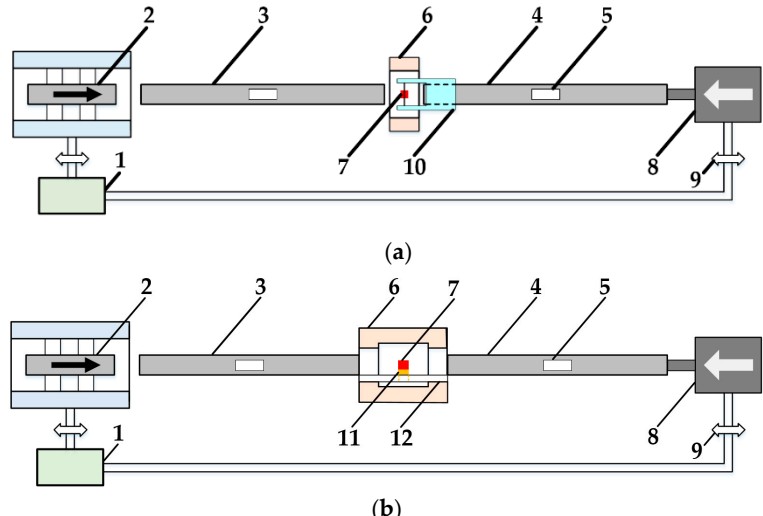

(**a**)

(**b**)

**Figure 2.** SHPB devices: (**a**) fine loading bars for Φ2 × 2 samples at high strain-rate; (**b**) coarse loading bars for Φ3 × 3 samples at low strain-rate. 1—air chamber; 2—strike bar; 3—incident bar; 4—transmission bar; 5—strain gauge; 6—furnace; 7—sample; 8—synchronous device; 9—valve; 10—clamping device; 11—heat insulation block; 12—guide rail.

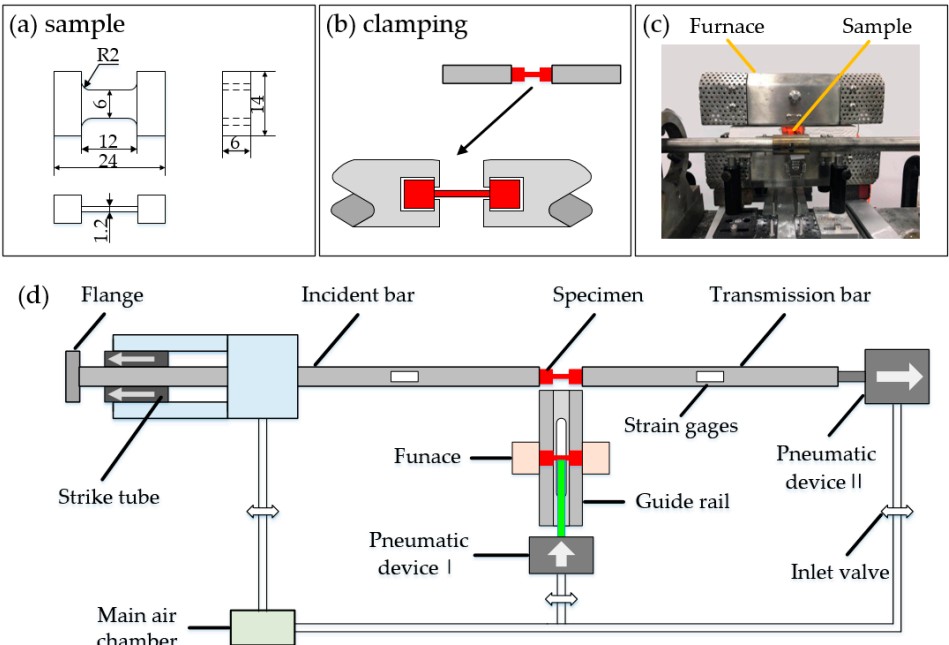

**Figure 3.** SHTB technique at high temperature with a double pneumatic device. (**a**) Samples used in the tests; (**b**) clamping device; (**c**) the real SHTB device used; (**d**) experimental device schematic.

The sample is deformed due to the impact load when the incident wave reaches the surface between the sample and incident bar, after which part of the incident wave converts into the transmitted wave that travels forward along the transmission bar, and another portion converts into the reflected wave that travels backward the incident bar.

### 2.3. Quasi-Static Tensile Tests with Notched Sample

The quasi-static tensile tests with notched samples are carried out at the strain rate of $10^{-3}$ for fitting the constants of J-C damage model, the notched samples are shown in Figure 4 with three notch radii of 3 mm, 4 mm and 5 mm for three specified values of triaxiality. According to the research of Bridgman [18], the triaxiality can be calculated by the following formula:

$$\sigma^* = \frac{1}{3} + \ln\left(1 + \frac{a}{2R}\right) \tag{3}$$

where $a$ and $R$ are the radius at the notched section and notch radius, respectively.

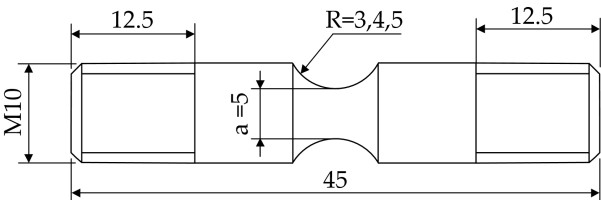

**Figure 4.** Notched sample for the specified triaxiality (units: mm).

### 2.4. Test Scheme

The $\Phi 2 \times 2$ cylindrical samples are prepared for high strain-rate SHPB tests at the strain rate higher than $3 \times 10^3$ s$^{-1}$, which are performed with the device in Figure 2a, and $\Phi 3 \times 3$ ones for a low strain-rate SHPB test at the strain rate lower than $3 \times 10^3$ s$^{-1}$, which are performed with the device in Figure 2b. The SHPB tests are performed under 12 different loading conditions with the specified combinations of strain rate and temperature. The strain rate are $10^3$ s$^{-1}$, $2 \times 10^3$ s$^{-1}$, $3 \times 10^3$ s$^{-1}$,

$5 \times 10^3$ s$^{-1}$, $8 \times 10^3$ s$^{-1}$ and the temperatures are 25 °C, 300 °C, 900 °C. The tests under each condition are repeated at least three times.

The dog-bone shape samples shown in Figure 3a are used for SHTB tests, and the tests are conducted at 1000 s$^{-1}$ (25 °C) and 25 °C, 300 °C, 600 °C, 750 °C, 900 °C (500 s$^{-1}$) respectively.

## 3. Result and Analysis

### 3.1. Mechanical Behavior Analysis

Figure 5a,b show the flow stress-plastic strain curves of 45CrNiMoVA from quasi-static compression tests and SHPB tests, respectively. The flow stress is obtained by offsetting the elastic stage of the true stress-true strain curve by 0.2%, and the plastic strain can be calculated as: $\varepsilon_\mathrm{p} = \varepsilon - \sigma/E$, where $\varepsilon$ is the true strain, $\sigma$ is the true stress and $E$ is the elastic modulus. The relationship of the flow stress to plastic strain, strain rate and temperature are studied on the basis of these curves.

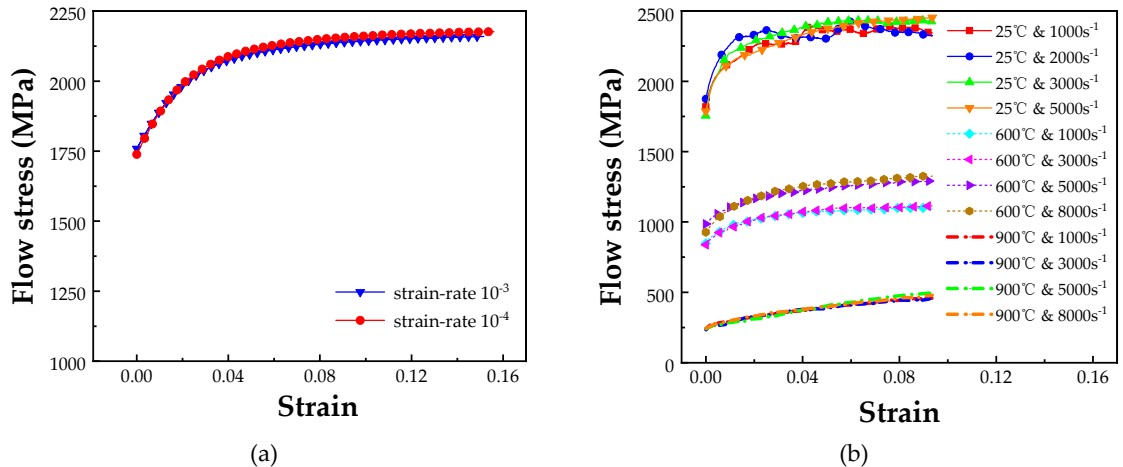

(a) (b)

**Figure 5.** (**a**) Quasi-static and (**b**) SHPB compression test results.

It is noted that in the quasi-static stage the strain hardening effect is not significant and the flow stress changes very little with the increasing of plastic strain as shown in Figure 5a, so the reference strain rate for fitting the constitutive model is set to $10^{-3}$.

Figure 5b from the SHPB tests shows the flow stress-plastic strain relation at different strain rate and temperature. Obviously, this material is characterized by significant thermal softening effect that the flow stress decreases rapidly with the increasing of temperature. But the strain hardening effect becomes complex and is temperature-related: at low temperature (<600 °C), the strain hardening effect is not significant; but at the temperature of 900 °C, the flow stress increases linearly with the plastic strain, and the strain hardening effect becomes significant. This phenomenon is considered to be related with the phase transition, when the temperature reaches and exceeds the critical phase transition temperature, some tempered martensite grains in the metal gradually transforms into austenite grains [19,20], resulting in the decrease of strength but the increase of plasticity.

The influence of strain rate hardening is evident for the different strain rates and temperatures upon which the tests were done; especially at temperatures below 600 °C. With the strain rate increasing, the time for the dislocation to overcome the barrier is reduced, and higher stress is required under the condition that the total energy required doesn't change, so the flow stress of the material becomes higher [21,22] and the strain rate hardening effect is especially high at 600 °C. While at higher temperature, the material plasticity is greatly improved. Here thermal softening is dominant, so the influence of strain rate hardening effect on the flow stress is suppressed.

*3.2. Plastic Constitutive Model Fitting*

The J-C constants (*A*, *B*, *C*, *n*, *m*) are firstly determined based on the quasi-static compression and SHPB test data. The most commonly used method to identify the J-C constants for metal is the linear regression strategy [23]. But in this paper, the multi-objective optimization fitting method proposed by Xiang is used for parameter fitting [9]. With this method, the data obtained from the quasi-static compression and SHPB compression tests can be used simultaneously, and the multi-variable coupling relationships of strain, strain rate and temperature are considered as well.

The fitting results of classic J-C constitutive model of 45CrNiMoVA ultrahigh strength steel in Figure 6 show that the J-C constitutive model can accurately describe the plastic behaviors of the material at room temperature for both quasi-static and high strain-rate. But when the temperature increases up to 600 °C, the fitted classic J-C model curve are obviously deviated from the experimental data, lower at 600 °C and higher at 900 °C than the experimental ones.

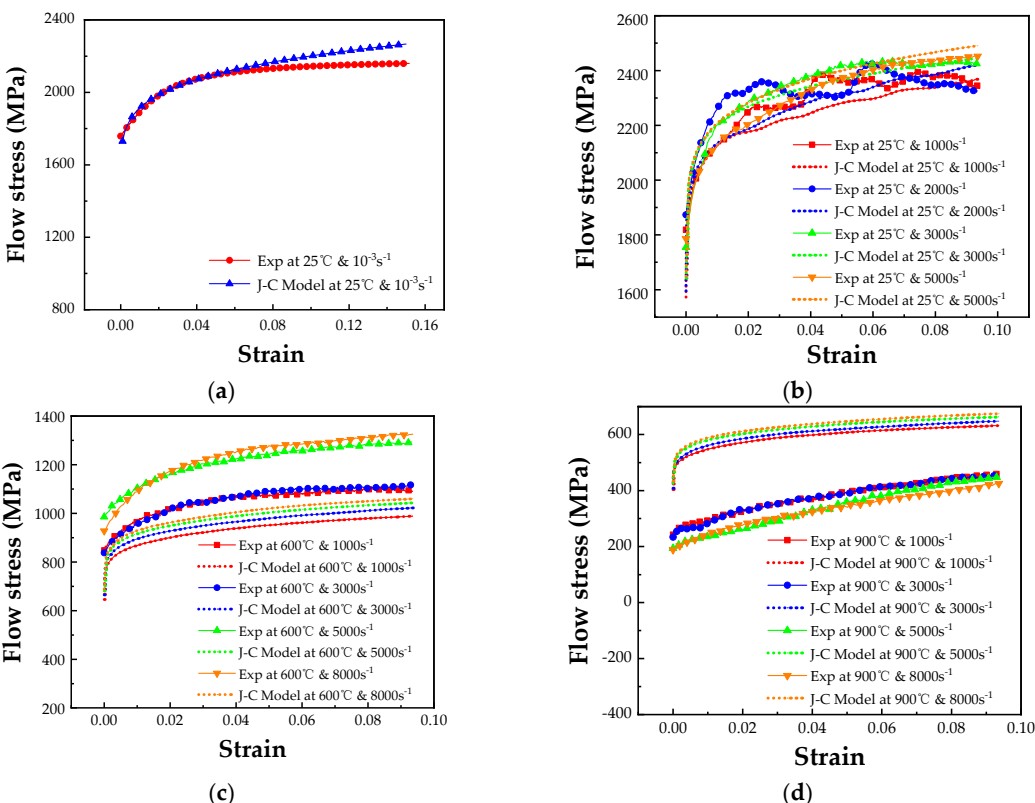

**Figure 6.** Comparison of experimental data and J-C model (**a**) quasi-static at 25 °C; (**b**) at different strain rate and 25 °C; (**c**) at different strain rate and 600 °C; (**d**) at different strain rate and 900 °C.

The fitting accuracy $R^2$ of the fitted classic J-C model reaches only 96.67%. According to the analysis in Section 3.1, as the temperature increases up to the critical temperature, the tempered martensite changes to the austenite phase. The great phase changes in material properties at high temperature cannot be reflected sufficiently by a simple temperature-related term $1 - (T^*)^m$. Therefore, it is necessary to modify the J-C constitutive model by considering the phase change of 45CrNiMoVA ultrahigh strength steel.

First, a temperature-dependent item $f(T)$ used to express the phase change is added to the J-C model; thus, the constitutive model becomes:

$$\sigma = (A + B\varepsilon_P{}^n) \times \left(1 + C\ln\left(\frac{\dot{\varepsilon}}{\dot{\varepsilon}_0}\right)\right) \times \left(1 - (T^*)^m\right) \times f(T) \tag{4}$$

$$f(T) = aT^3 + bT^2 + cT + d \tag{5}$$

where a, b, c, d are temperature-related constants, and a $= -4.086 \times 10^{-9}$, b $= 3.07 \times 10^{-6}$, c $= -3.963 \times 10^{-5}$, d $= 1.031$. Compared with the experimental data at different temperatures, the modified J-C model is consistent with the experimental curves, and the $R^2$ value is 98.91%, which means this model is more accurate.

Furthermore, in order to reflect the strong strain hardening phenomenon of the material at the temperature of 900 °C, a strain-temperature coupled term $g(T, \varepsilon)$ is added to the modified model for higher fitting accuracy, and the J-C constitutive model is finally modified to:

$$\sigma = (A + B\varepsilon_P{}^n) \times \left(1 + C\ln\left(\frac{\dot{\varepsilon}}{\dot{\varepsilon_0}}\right)\right) \times (1 - (T^*)^m) \times f(T) + g(T, \varepsilon) \tag{6}$$

$$g(T, \varepsilon) = k_t \times k_\varepsilon \times e^{(-T_c{}^2 - \varepsilon_c{}^2)} \tag{7}$$

$$T_c = \frac{T - p_t}{q_t} , \ \varepsilon_c = \frac{\varepsilon - p_\varepsilon}{q_\varepsilon} \tag{8}$$

where $k_t$ and $k_\varepsilon$ is the temperature coupling coefficient and the strain coupling coefficient, respectively, $T_c$ and $\varepsilon_c$ is the coupling temperature and coupling strain. For 45CrNiMoVA material, $k_t = 1.002$, $p_t = 888.3$, $q_t = 93.6$; $k_\varepsilon = 222.5$, $p_\varepsilon = 0.1627$, $and$ $q_\varepsilon = 0.105$. By adding this coupled term, the fitted curves at room temperature and 600 °C are changed to some extent, but the strain hardening effect at 900 °C is perfectly taken into account, and the fitting accuracy $R^2$ can reach as high as 99.53%. By comparing Figure 6 with Figure 7, a high degree of fitting between the modified J-C model and the experimental data is well manifested. But the fitted parameters in Table 4 shows very little changes from the J-C model to the modified J-C model, so the increase in fitting accuracy is owed to the added two terms.

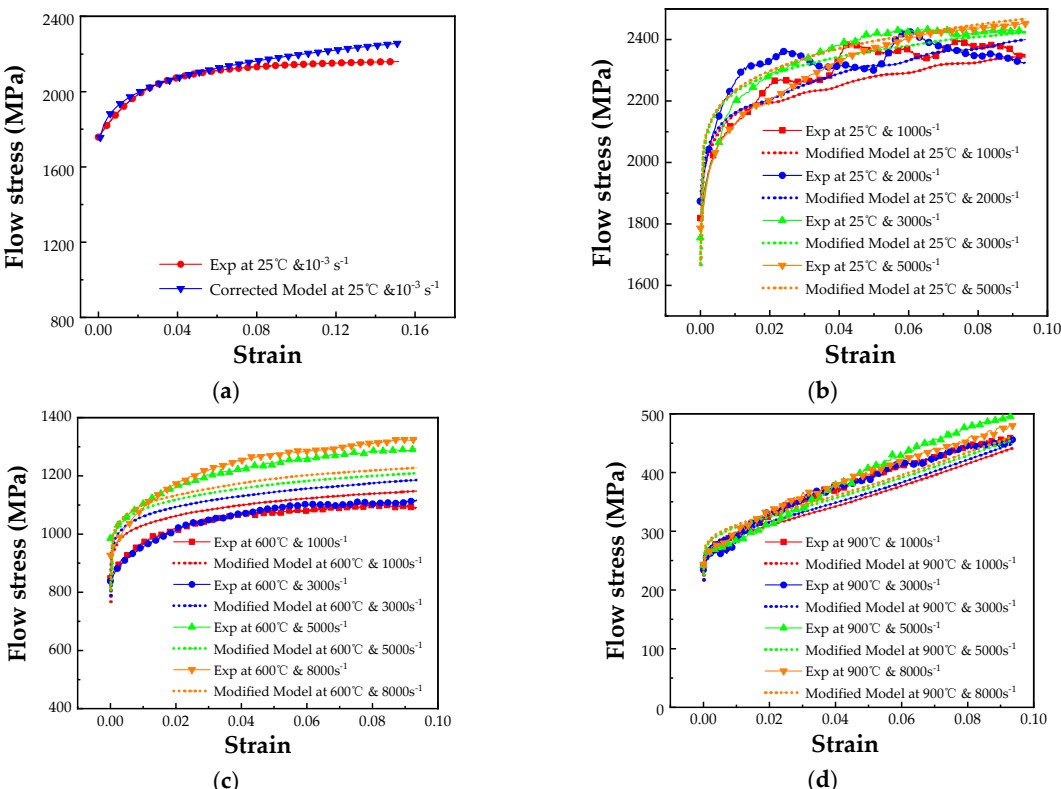

**Figure 7.** Comparison of experimental data and modified J-C model (**a**) quasi-static at 25 °C; (**b**) at different strain rate and 25 °C; (**c**) at different strain rate and 600 °C; (**d**) at different strain rate and 900 °C.

| Models | $A$ | $B$ | $C$ | $n$ | $m$ | $R^2$ |
|---|---|---|---|---|---|---|
| J-C model | 1404 | 1247 | 0.009101 | 0.1943 | 0.5724 | 96.67% |
| Modified J-C model | 1410 | 1124 | 0.008786 | 0.1954 | 0.5622 | 99.53% |

*3.3. Johnson-Cook Damage Model Constants Fitting*

According to the cumulative damage rules, the material fails until the damage parameter $D$ exceeds 1, and $D$ is determined as:

$$D = \sum \frac{\Delta \varepsilon_{eq}}{\varepsilon_f} \tag{9}$$

where $\Delta \varepsilon_{eq}$ is an equivalent plastic strain increment during an integral cycle and $\varepsilon_f$ is the effective fracture strain at the current time step. As the most commonly used damage model, J-C damage model takes into account the influence of triaxiality, strain rate and temperature on the effective fracture strain, and the effective fracture strain is expressed as [3]:

$$\varepsilon_f = \left(D_1 + D_2 \exp(D_3\, \sigma^*)\right) \times \left(1 + D_4 \times \ln\left(\frac{\dot{\varepsilon}_{eq}}{\dot{\varepsilon}_0}\right)\right) \times (1 + D_5 T^*) \tag{10}$$

where $\sigma^* = \sigma_m/\sigma_e$ is the stress triaxiality, $\sigma_m = (\sigma_1 + \sigma_2 + \sigma_3)/3$ is mean stress, and $\sigma_e = \frac{\sqrt{2}}{2} \times \sqrt{(\sigma_1 - \sigma_2)^2 + (\sigma_2 - \sigma_3)^2 + (\sigma_1 - \sigma_3)^2}$ is the Von Mises equivalent stress, $\sigma_1$, $\sigma_2$, $\sigma_3$ are the principal stresses; $\dot{\varepsilon}_{eq}$ is the equivalent strain rate, and $\dot{\varepsilon}_0$ the reference strain rate, which is usually set to $10^{-3}$; $T^*$ is calculated as in Equation (2).

There are five J-C damage constants $D_1$, $D_2$, $D_3$, $D_4$, $D_5$, which will be identified by experiment. $D_1$, $D_2$, $D_3$ are stress triaxiality-related, $D_4$ is strain rate related and $D_5$ temperature related. The SHTB tests are performed at different strain rate and temperature for fitting $D_4$ and $D_5$, and the quasi-static notched tensile tests are carried out for fitting $D_1$, $D_2$, $D_3$ parameters.

Table 5 lists the notch radius R of the samples, the initial cross-sectional area $A_0$ space and final fracture cross-sectional area $A_f$, the calculated stress triaxiality $\sigma^*$ by Equation 3 and the fracture strain $\varepsilon_f$ which is calculated as [24]:

$$\varepsilon_f = \ln\left(A_0/A_f\right) \tag{11}$$

**Table 5.** Tensile samples measurement results.

| Sample Numbers | A | B | C | D | E |
|---|---|---|---|---|---|
| Notch Radius (mm) | 3.0 | 4.0 | 5.0 | 0.0 | 0.0 |
| Initial Section Area $A_0$ (mm$^2$) | 19.6350 | 19.6350 | 19.6350 | 19.6350 | 19.6350 |
| Section Area after Fracture $A_f$ (mm$^2$) | 8.3251 | 8.2091 | 8.0968 | 5.5641 | 5.3778 |
| Fracture Strain $\varepsilon_f$ | 0.8580 | 0.8721 | 0.8858 | 1.2610 | 1.2950 |
| Triaxiality $\sigma^*$ | 0.6816 | 0.6053 | 0.5565 | 0.3333 | 0.3333 |

A three-step fitting process is implemented for fitting the damage model constants. The damage constants in the three items of J-C damage model are fitted by using the linear regression algorithm step by step. In each fitting step, only one single variable of triaxiality, strain rate and temperature is changed and the two others are kept constant. The three-step fitting process is performed as follows:

Step 1: For quasi-static tensile and notch sample tensile test at room temperature, the strain rate related term and the temperature related term in J-C damage model can be omitted. After logarithmic transformation, J-C damage model is given to Equation (12). Based on the data points of $\ln\left(\varepsilon_f - D_1\right)$ against triaxiality $\sigma^*$ from quasi-static tensile and notch sample tensile tests at room temperature,

$D_1$, $D_2$ and $D_3$ in the stress triaxiality related term are fitted. The fitting result is shown in Figure 8a, and the fitted values are: $D_1 = 0.805$; $D_2 = 5.002$; $D_3 = -7.095$, with a fitting precision of 98.95%.

$$\ln\left(\varepsilon_f - D_1\right) = D_3\,\sigma^* + \ln(D_2) \tag{12}$$

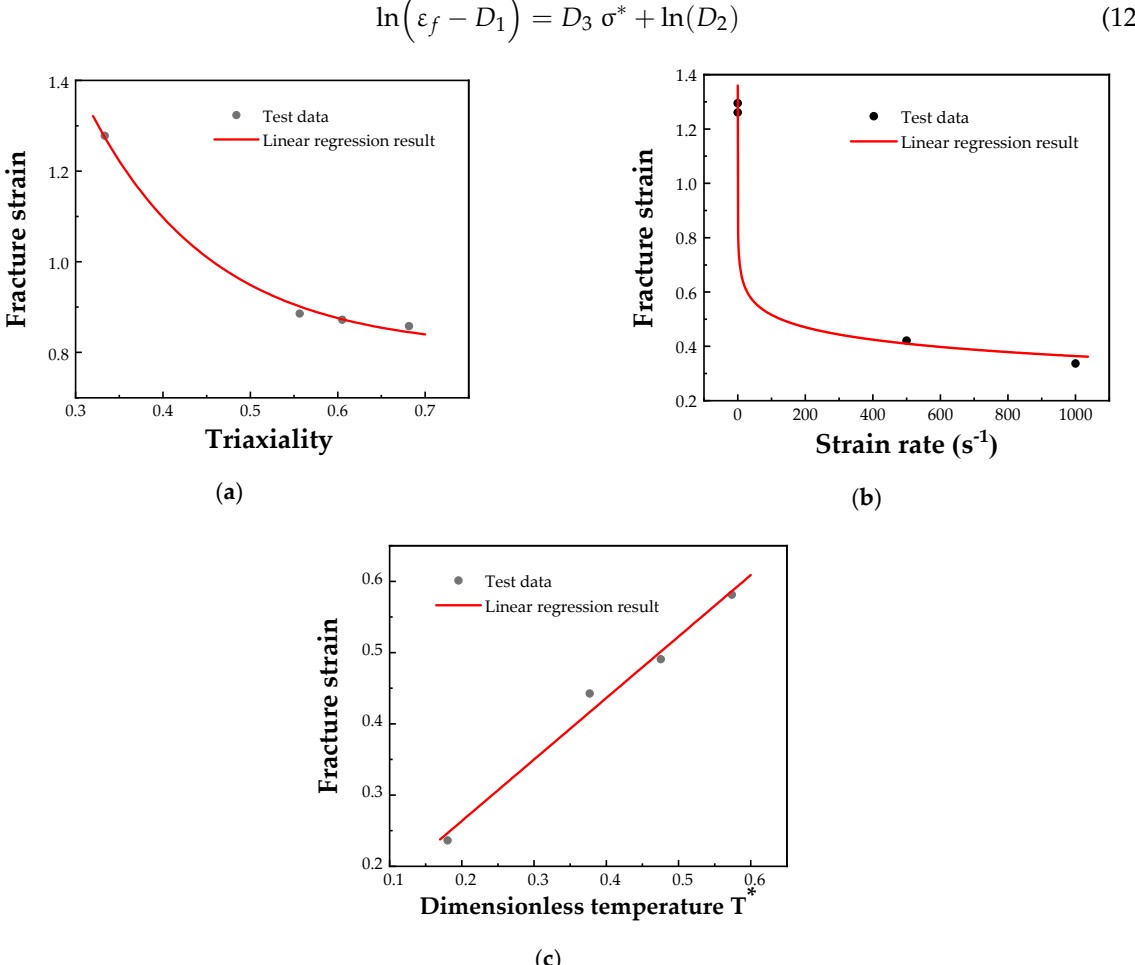

**Figure 8.** J-C damage model constants fitting results (**a**) triaxiality related constants $D_1$, $D_2$, $D_3$; (**b**) strain rate related constants $D_4$; (**c**) temperature related constants $D_5$.

Step 2: For SHTB test at room temperature, the temperature related term in J-C damage model can be omitted. The J-C damage model is transformed to Equation (13). Based on the data points of $\varepsilon_f / (D_1 + D_2 \exp(D_3\,\sigma^*))$ against $\ln(\dot{\varepsilon}_{eq}/\dot{\varepsilon}_0)$ from SHTB at room temperature, $D_4$ in the strain rate related term is fitted. Fitting results are shown in Figure 8b and the value of $D_4$ is $-0.140$, with a fitting precision of 95.17%. The relationship for $D_4$ is given below:

$$\frac{\varepsilon_f}{(D_1 + D_2 \exp(D_3\,\sigma^*))} = D_4 \times \ln\left(\frac{\dot{\varepsilon}_{eq}}{\dot{\varepsilon}_0}\right) + 1 \tag{13}$$

Step 3: Obtain the value of $D_5$ through the linear regression of $\varepsilon_f$ against $T^*$ under the same strain rate in SHTB test. As shown in Figure 8c, the value of $D_5$ is equal to the ratio of the slope of the fitted curve to its intercept, and its value is 9.470 with a fitting precision of 98.56%.

$$\frac{\varepsilon_f}{(D_1 + D_2 \exp(D_3\,\sigma^*)) \times \left(1 + D_4 \times \ln\left(\frac{\dot{\varepsilon}_{eq}}{\dot{\varepsilon}_0}\right)\right)} = D_5 T^* + 1 \tag{14}$$

## 4. Finite Element Simulation Verification

In order to verify the modified J-C constitutive model and J-C damage model, finite element method simulations are performed. ABAQUS finite element software (6.14, Dassault Simulia, Paris, France) provides users with many subroutine interfaces, allowing users to extend the main program's functions and develop their own models including geometric models, materials and loads, etc. In addition to the commonly used explicit VUMAT and implicit UMAT material subroutines, the VUHARD subroutine is often used to define the constitutive model of isotropic materials. The subroutine implements real-time data exchange with the ABAQUS main program. The main program provides variables such as stress, strain, strain rate, temperature, etc. to the subroutine, and the data is returned to the main program after stress updating by subroutine. The VUHARD subroutine is developed based on the modified J-C constitutive model for the finite element simulation of tensile and compression process of 45CrNiMoVA ultrahigh strength steel.

The dynamic, temperature-displacement, explicit step and the 8-node thermally coupled brick, trilinear displacement and temperature (C3D8T) elements are used for both SHPB and SHTB finite element simulations. As shown in Figure 9, the SHPB simulation model consists of two $\Phi 5 \times 1250$ mm loading bars and a sample, and the transmission bar is fixed at one end and the impact of the strike bar on incident bar is simplified as an impact load similar to the tested actual incident wave in Figure 10. As for the SHTB simulation, the dog-bone shape sample is fixed at one end and the impact load is directly loaded on the other end. The temperature of samples are set to 25 °C, 600 °C and 900 °C respectively at the initial step. The size of the minimum elements in SHPB simulation is set to $0.2 \times 0.2 \times 0.2$ mm$^3$, but the samples in SHTB simulation are discretized with much finer elements ($0.06 \times 0.06 \times 0.06$ mm$^3$) for smooth fracture. In addition, the size of element has little effect on the magnitude and distribution of stress, temperature, etc., but the fracture depends on it, and the displacement at failure is set as 0.5 times of the finest element. The specific heat, thermal expansion coefficient and thermal conductivity are assumed as temperature-independent and keep constant as given in Table 2.

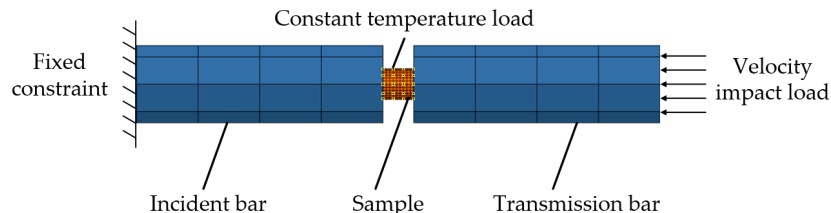

**Figure 9.** SHPB simulation model.

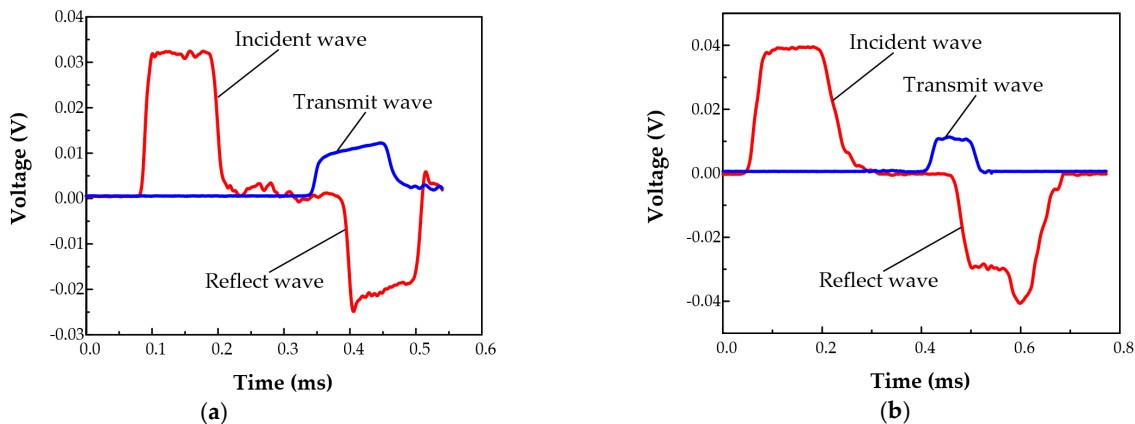

**Figure 10.** (**a**) Original wave signals in SHPB test; (**b**) original wave signals in SHTB test.

In order to verify the modified J-C constitutive model and the J-C damage model, the stress-strain curves from the SHPB simulations and the fracture morphology of the samples from the SHTB

simulations are compared with the ones from the corresponding tests. Figure 11 demonstrates the effect of temperature on the fracture morphology at the tensile strain-rate of 1000 s$^{-1}$. On the samples after the SHTB tests from 25 to 600 °C, X-shaped shear band are observed, and a V-shaped fracture form is formed along the shear band or a fracture plane tilted to the loading direction at an approximate angle of 45° is formed due to the lateral load component caused by the test clamping error. As the temperature rises above 750 °C, intense oxidation, thermal expansion and obvious elongation of the samples are observed. The fracture surface is perpendicular to the loading direction, and obvious necking can be observed due to significant thermal softening. Figure 12 shows the flow stress–strain curves from the SHPB simulations and tests, and the simulation results are in good agreement with the test results.

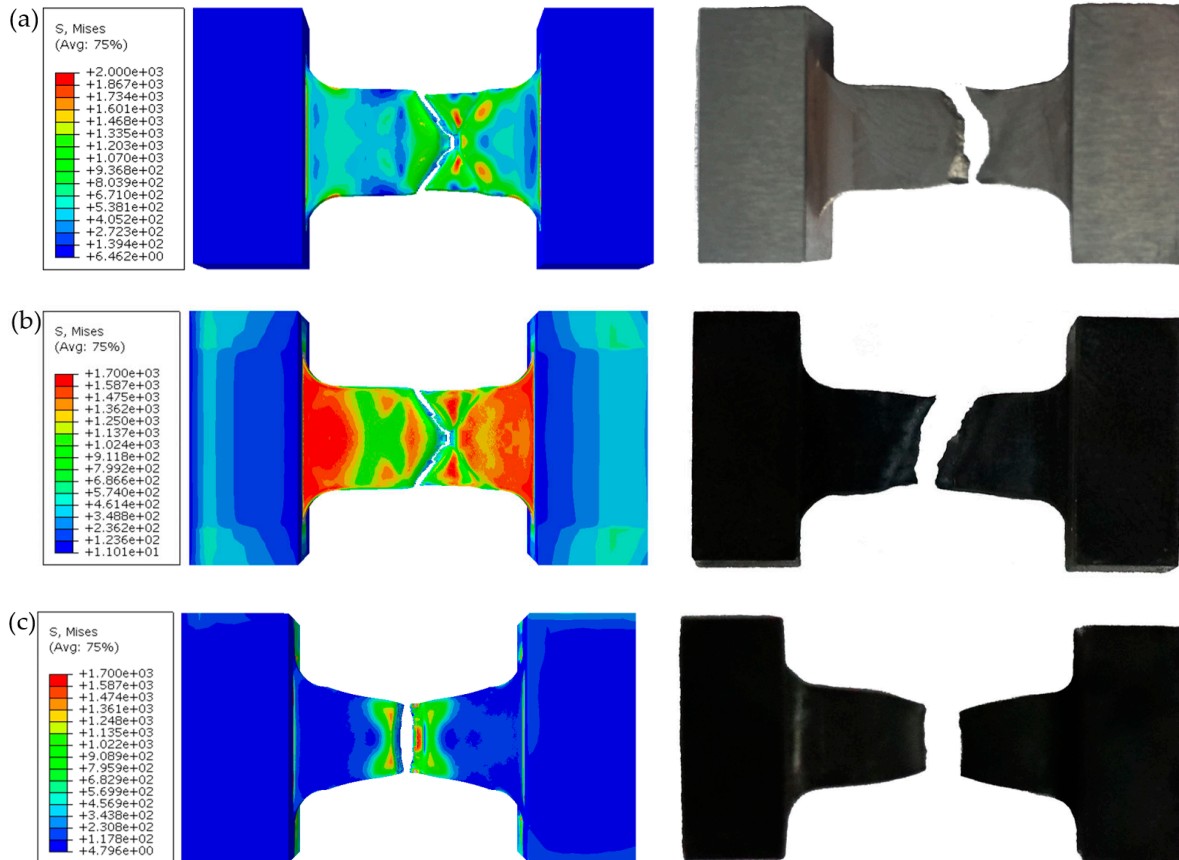

**Figure 11.** Fracture morphology in SHTB test and simulation at (**a**) 25 °C; (**b**) 600 °C; and (**c**) 900 °C.

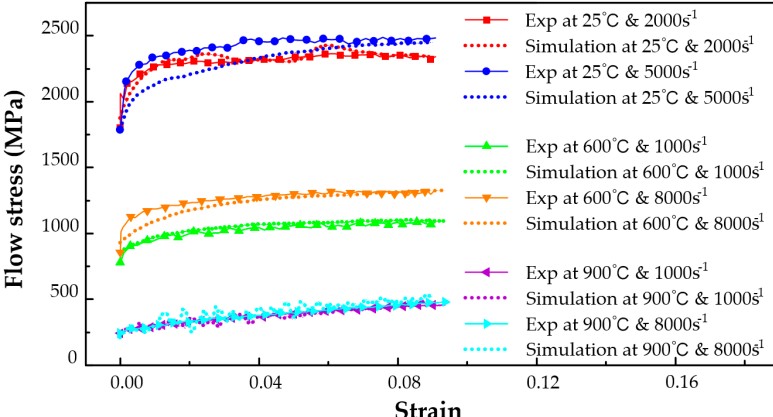

**Figure 12.** Flow stress-strain curves of SHPB tests and simulation at different strain rate and temperature.

## 5. Conclusions

In this paper, dynamic and quasi-static compression and tensile tests are implemented to study the mechanical properties of 45CrNiMoVA ultra-high-strength torsion bar steel. The effect of strain, strain rate and temperature on flow stress is studied based on the flow stress-strain curves obtained through the improved SHPB and SHTB tests at high strain-rate and temperature. In addition, a modified J-C constitutive model is established and the J-C damage model constants are fitted.

(1) According to the improved SHTB and SHPB tests, 45CrNiMoVA steel is characterized by obvious temperature softening effect, the yield strength and flow stress decreases significantly with increasing temperature. The obvious strain-rate hardening effect at 600 °C is considered to be related to the activated dislocations rephrase by temperature. And, the especially significant strain hardening effect at 900 °C is assumed to result from the phase transition from initial tempered martensite to austenite.

(2) In order to describe the special temperature softening and strain-hardening effect at high temperature, a modified J-C model is established by incorporating a temperature-related term and a strain-temperature coupled term. Based on the multi-objective optimization fitting strategy, the model parameters are fitted with a fitting accuracy of 99.53% in $R^2$. The constants of J-C damage model are fitted though a three-step fitting process based on the SHTB tests and quasi-static tensile tests with notched samples.

(3) The stress-thermal coupled finite element simulations of SHPB and SHTB tests at different strain rates and temperature are conducted using ABAQUS. A user subroutine called VUHARD, based on the modified J-C constitutive model, is developed and incorporated into the simulations. The fracture morphology and flow stress-strain from the tests and the simulations are compared, and their good agreement means that the modified constitutive model and damage model constants are quite suitable for this material.

**Author Contributions:** Methodology and experimental design, X.H., L.X. and F.G.; performing of experiments, X.H., L.X. and F.G.; data processing, fitting of parameters and establishment of constitutive model, X.H. and L.X. and J.X.; simulation verification, X.H., L.X. and F.G.; paper-writing and editing, X.H. and L.X.

**Funding:** This research was funded by the National Natural Science Foundation of China under Grant No. 51575051.

**Acknowledgments:** The authors are grateful for the technical support from the School of Aeronautics at Northwestern Polytechnical University.

**Conflicts of Interest:** The authors declare no conflicts of interest.

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
