# Peer review of "On the Development of Material Constitutive Model for 45CrNiMoVA Ultra-High-Strength Steel"

_metals, doi:10.3390/met9030374_

Round 1

Reviewer 1 Report

Dear authors,

   overall the paper is well structured with interesting scientific results. Changes to the wording and the language used need to be made. Below are my main comments for the manuscript. Attached you may find the detailed corrections I implemented to the manuscript.

Terminology used is sometimes not appropriate and needs to be changed; please refer to the document attached.

In certain instances the language in the text and the style needs to be edited as well.

The authors present only the final heat treatment; they would need to include the whole process history to understand how the material is manufactured. It is true that the final tempering process is crucial as to the final mechanical properties in the part. Yet also previous manufacturing steps are equally crucial and important; especially with respect to the embedded stresses in the part; the way the part fractures upon tensile testing etc.

In figure 5 the authors use similar colours for different testing configurations. They need to change the colours on the plot to make it more distinguishable between the different curves.

The wording in sentences/rows 175-177 is not appropriate and conflicting with previous statements; thus it needs to be modified.

Equation 9 needs to be rewritten

Kind regards.

Author Response

Response to Reviewer 1 Comments

Point 1: Terminology used is sometimes not appropriate and needs to be changed; please refer to the document attached. In certain instances the language in the text and the style needs to be edited as well.

Response 1: Thanks for your suggestions, the terminology mentioned in the document and some other expression errors have been changed.

Point 2: The authors present only the final heat treatment; they would need to include the whole process history to understand how the material is manufactured. It is true that the final tempering process is crucial as to the final mechanical properties in the part. Yet also previous manufacturing steps are equally crucial and important; especially with respect to the embedded stresses in the part; the way the part fractures upon tensile testing etc.

Response 2: The investigated 45CrNiMoVA low carbon alloy steel has experienced 2.5-hour annealing at 920℃, 1.5-hour quenching at 780℃ and 4-hour low temperature tempering at 200℃.

Point 3: In figure 5 the authors use similar colours for different testing configurations. They need to change the colours on the plot to make it more distinguishable between the different curves.

Response 3: The lines and colours in the Figure 5 have been adjusted, and it becomes more distinguishable.

Point 4: The wording in sentences/rows 175-177 is not appropriate and conflicting with previous statements; thus it needs to be modified.

Response 4: The sentences in Lines 175-177 have been modified, statements without too much data support have been deleted, only strength is mentioned. 

Point 5: Equation 9 needs to be rewritten.

Response 5: The equation (9) has been modified. 

Reviewer 2 Report

I propose to fix the equation (2). It should have the shape of the equation.

Figure 1 is poorly readable.

The values in Tab. 2 are measured for a specific material, or are they taken from literature?

Have all the strain rates and temperatures achieved the same deformation? (Figure5,6,7, 12).

The article is interesting. It solves the problem of the behavior of the steel (45CrNiMoVA) under tension and compressure at high strain rates and temperatures up to 900 ° C.

Author Response

Response to Reviewer 2 Comments

Point 1: I propose to fix the equation (2). It should have the shape of the equation.

Response 1: The equation has been modified accordingly. 

Point 2: Figure 1 is poorly readable.

Response 2: Figure 1 have been adjusted, more pictures and additional descriptions are added in Figure 1 and in the text. As shown in Figure 1: (a) sample directions: RD represents rolling-direction, TD represents transverse-direction and ND represents normal-direction; (b) SEM metallographic graph; (c) distribution of martensite and austenite grians, geen area represents tempered martensite and red area represents retained austenite; (d) RD inverse pole figure; (e) color code map of the RD inverse pole figure.

Point 3: The values in Tab. 2 are measured for a specific material, or are they taken from literature? 

Response 3: Most of the values in table 2 are taken from literature, such as density, melting point, possion’s ratio, specific heat, thermal expansion coefficient and thermal conductivity. The source is added in the reference.

The elastic modulus and reference strain-rate are measured from quasi-static compression and tensile tests.

Point 4: Have all the strain rates and temperatures achieved the same deformation? (Figure5,6,7, 12).

Response 4: For quasi-static compression tests, all the samples were compressed to the strain of 0.5. For quasi-static tensile tests, all the samples were pulled off. For SHPB tests, samples under 25℃, strain-rate of 2000s-1 and 5000s-1 loading conditions were fractured; part of the samples under 600℃, strain-rate of 5000s-1 loading condition and all the samples under 600℃, strain-rate of 8000s-1 loading condition were fractured; no samples under 900℃ loading conditions were fractured. For SHTB tests, except that two 900 °C samples were not fractured, the other samples under all loading conditions were fractured. 

Reviewer 3 Report

The paper presents the development of a modified Johnson-Cook model for the mechanical behavior of an Ultra High Strength Steel.

It is globally sound and well written. A few comments and improvements are however required as listed  below:

line 43-49: explicit acronyms (SHPB) in the text at their first occurrence.

lines 81-85: sounds already like a conclusion instead of an introduction to the following work.

line 90, Fig 1: add specimen directions in the maps and specify what type of EBSD fig 1.b) is: Rolling-Direction Inverse Pole Figure map ? Add the corresponding color code.

lines 112-125: rather difficult to read, some of the sentences should be split / made simpler.

line 239: problem in equation (9)

line 246: the Mises stress expression given assumes sigma1 to sigma3 are the principal stresses. Specify it or give the general expression including shear components

Table 5: Stress triaxiality in tension should positive

Section 4: details are missing for the reader to be able to reproduce this type of FE simulation:

- what is the sample mesh size, how do the presented results (e.g. stress field and fracture) depend on it ?

- Fig 9 should be enhanced with additional details / sketches of the boundary conditions.

THe temperature boundary conditions are not detailed at all, they must be included. Constant temperature assumed through the analysis ?

- There are no other details on the thermophysical properties than the one listed in Table 2. Does it mean that these numbers are assumed as not depending on Temperature ? If yes it must be specified.

Fig 11: increase the size of the color scale legend. Barely readable here.

Author Response

Response to Reviewer 3 Comments

Point 1: line 43-49: explicit acronyms (SHPB) in the text at their first occurrence.

Response 1: The explicit acronyms SHPB in this paper is first proposed in abstract lines 14-15.

Point 2: lines 81-85: sounds already like a conclusion instead of an introduction to the following work.

Response 2: I agree to the suggestion of reviewer, the content of Line 81-85 has been concluded in the conclusion and seems repeated when it appears here, so Line 81-85 has been deleted.

Point 3: line 90, Fig 1: add specimen directions in the maps and specify what type of EBSD fig 1.b) is: Rolling-Direction Inverse Pole Figure map? Add the corresponding color code.

Response 3: The specimen direction is added in Fig. 1, the sample is a 10×10×10 cube and the observed surface is the surface perpendicular to the RD direction. Fig. (b) is the inverse pole figure in the Rolling-Direction and the additional description is added in the text. The corresponding color coded map is added in Fig 1.

Point 4: lines 112-125: rather difficult to read, some of the sentences should be split/made simpler.

Response 4: The sentences in Lines 112-125 are simplified accordingly.

Point 5: line 239: problem in equation (9)

Response 5: The equation (9) has been modified.

Point 6: line 246: the Mises stress expression given assumes sigma1 to sigma3 are the principal stresses. Specify it or give the general expression including shear components.

Response 6: It has been specified that  Ïƒ1, σ2, σ3 are the principle stresses in Lines 248.

Point 7: Table 5: Stress triaxiality in tension should positive.

Response 7: Thanks for the carefulness of reviewer. The stress triaxiality for tensile samples in Table 5 have been corrected into positive value, and corresponding value of D3 and Fig. 8 (a) have been changed as well.

Point 8: Section 4: details are missing for the reader to be able to reproduce this type of FE simulation:

- what is the sample mesh size, how do the presented results (e.g. stress field and fracture) depend on it ?

- Fig 9 should be enhanced with additional details / sketches of the boundary conditions.

The temperature boundary conditions are not detailed at all, they must be included. Constant temperature assumed through the analysis ?

- There are no other details on the thermophysical properties than the one listed in Table 2. Does it mean that these numbers are assumed as not depending on Temperature ? If yes it must be specified.

Response 8: Details in the simulation about mesh size (0.2×0.2×0.2mm elements in SHPB simulation and 0.06×0.06×0.06mm elements in SHTB simulation) is added in the text, and the effect of elements size has influence on the fracture, but little influence on the magnitude and distribution of stress, temperature.

The boundary conditions, speed load and temperature load are added in Figure 9, and the temperature is kept constant in the simulation.

The influence of temperature on specific heat, thermal expansion coefficient and thermal conductivity are not considered in the simulation and is specified in the revision.

Point 9: Fig 11: increase the size of the color scale legend. Barely readable here.

Response 9: The color scale legend in Fig 11 is magnified.

Round 2

Reviewer 3 Report

The manuscript has been improved according to the comments and also checked for English mistakes. A few details below:

INtroduction: please specify the acronyms not only in the abstract but also the 1rst time used in the main text.

Fig 1: since the 2 phases use the same IPF color code, one color-code triangle is enough

l270: principal stresses (not "principle")

Author Response

Response to Reviewer 3 Comments

Point 1: Introduction: please specify the acronyms not only in the abstract but also the 1rst time used in the main text. 

Response 1: The explicit acronyms SHPB in line 44 and SHTB in line 81 are specified, and the explicit acronyms J-C is modified too.

Point 2: Fig 1: since the 2 phases use the same IPF color code, one color-code triangle is enough 

Response 2: The Figure 1 (e) has been modified and only one color-code left.

Point 3: l270: principal stresses (not "principle") 

Response 3: The spelling mistake in Line 270 has been corrected.